 RESEARCH ADVANCE

# Structural screens identify candidate human homologs of insect chemoreceptors and cryptic *Drosophila* gustatory receptor-like proteins

**Richard Benton\*†, Nathaniel J Himmel\*†**

Center for Integrative Genomics, Faculty of Biology and Medicine, University of Lausanne, Lausanne, Switzerland

**Abstract** Insect odorant receptors and gustatory receptors define a superfamily of seven transmembrane domain ion channels (referred to here as 7TMICs), with homologs identified across Animalia except Chordata. Previously, we used sequence-based screening methods to reveal conservation of this family in unicellular eukaryotes and plants (DUF3537 proteins) (Benton et al., 2020). Here, we combine three-dimensional structure-based screening, ab initio protein folding predictions, phylogenetics, and expression analyses to characterize additional candidate homologs with tertiary but little or no primary structural similarity to known 7TMICs, including proteins in disease-causing *Trypanosoma*. Unexpectedly, we identify structural similarity between 7TMICs and PHTF proteins, a deeply conserved family of unknown function, whose human orthologs display enriched expression in testis, cerebellum, and muscle. We also discover divergent groups of 7TMICs in insects, which we term the gustatory receptor-like (Grl) proteins. Several *Drosophila melanogaster Grl*s display selective expression in subsets of taste neurons, suggesting that they are previously unrecognized insect chemoreceptors. Although we cannot exclude the possibility of remarkable structural convergence, our findings support the origin of 7TMICs in a eukaryotic common ancestor, counter previous assumptions of complete loss of 7TMICs in Chordata, and highlight the extreme evolvability of this protein fold, which likely underlies its functional diversification in different cellular contexts.

## Editor's evaluation

This article provides fundamental advances to our understanding of the ancestry of insect gustatory and olfactory receptors. It identifies new members of these two related ion channel families in distant species, and the strength of evidence is exceptional. This work will serve as a reference for scientists working on insect olfaction and for those working on molecular evolution

## Introduction

The insect chemosensory receptor repertoires of odorant receptors (Ors) and gustatory receptors (Grs) define a highly divergent family of ligand-gated ion channels, which underlie these animals' ability to respond to chemical cues in the external world (*Benton, 2015*; *Joseph and Carlson, 2015*; *Robertson, 2019*). Despite its vast size and functional importance, this family has long been an evolutionary enigma, displaying no resemblance to other classes of ion channels. Indeed, for many years, insect Ors and Grs were thought to be an invertebrate-specific protein class (*Benton, 2006*; *Robertson et al., 2003*). This view changed in the past decade, with the sequencing of a large number of genomes enabling the identification of homologs across animals (generally termed Gr-like [GRL]

**\*For correspondence:**
nathanieljohn.himmel@unil.ch
(NJH);
Richard.Benton@unil.ch (RB)

†These authors contributed
equally to this work

**Competing interest:** The authors
declare that no competing
interests exist.

**Reviewing Editor:** Claude
Desplan, New York University,
United States

proteins), including non-Bilateria (e.g., the sea anemone *Nematostella vectensis*), Hemichordata (e.g., the sea acorn *Saccoglossus kowalevskii*), various unicellular eukaryotes (e.g., the chytrid fungus *Spizellomyces punctatus* and the alga *Vitrella brassicaformis*) and Plantae (known as Domain of Unknown Function [DUF] 3537 proteins) (*Benton, 2015*; *Benton et al., 2020*; *Robertson, 2015*; *Saina et al., 2015*). For simplicity in nomenclature, we term here this broader superfamily (i.e., Ors, Grs, GRLs, and DUF3537 proteins) as 'seven transmembrane domain ion channels' (7TMICs), to distinguish them from unrelated 7TM G protein-coupled receptors. (We acknowledge that in most cases we do not know yet whether they are ion channels, and leave open the possibility for future updates to nomenclature.) Despite extensive searching, 7TMIC homologs have not been identified in Chordata, leading to proposals that these proteins were lost at or near the base of the chordate lineage (*Benton, 2015*; *Robertson, 2015*; *Saina et al., 2015*).

A substantial challenge in identifying 7TMIC homologs is their extreme sequence divergence (as little as 8% amino acid identity). The inclusion of proteins in this family relies primarily on the presence of topological features, notably seven TM domains and an intracellular N-terminus (*Benton et al., 2020*; *Benton et al., 2006*). Although insect Grs were originally recognized as possessing a short, conserved motif in transmembrane domain 7 (TM7) (described below) (*Robertson, 2019*; *Scott et al., 2001*), this motif is only partially or not at all conserved outside insects (*Benton et al., 2020*). For many protein families, the tertiary (three-dimensional) structure is generally more conserved than primary structure (*Illergård et al., 2009*; *Murzin et al., 1995*), and this property can offer an orthogonal strategy to identify homologous proteins. For the 7TMIC superfamily, the recent cryo-electronic microscopic (cryo-EM) structures of homotetrameric complexes of insect Ors (*Butterwick et al., 2018*; *Del Mármol et al., 2021*) provide important experimental insight into the tertiary structure of these proteins (as well as mechanistic insights into how these ion channels function). In our previous study (*Benton et al., 2020*), we used ab initio structural predictions of candidate 7TMIC sequences to reinforce our proposals of homology despite extremely low amino acid identity.

The recent breakthroughs in accuracy (to atomic level) and speed (seconds-to-minutes per sequence) of protein structure predictions, notably by AlphaFold2 (*Jumper et al., 2021*; *Varadi et al., 2022*), have now enabled millions of protein models to be generated. Here, we have exploited the unprecedented resource of the AlphaFold Protein Structure Database (*Jumper et al., 2021*; *Varadi et al., 2022*) and the Dali protein structure comparison algorithm (*Holm, 2022*), to screen for additional 7TMIC homologs by virtue of their tertiary structural similarity to experimentally determined insect Or structures.

## Results and discussion
### Tertiary structure-based screening for candidate 7TMIC homologs
Cryo-EM structures of two insect Ors have been obtained: the fig wasp (*Apocrypta bakeri*) Or co-receptor (Orco) (*Butterwick et al., 2018*; *Figure 1A–B*), which is a highly conserved member of the repertoire across most insect species (*Benton et al., 2006*; *Jones et al., 2005*; *Larsson et al., 2004*) and MhOr5 from the jumping bristletail (*Machilis hrabei*), a broadly tuned volatile sensor (*Del Mármol et al., 2021*). Despite sharing only 18% amino acid identity, these proteins adopt a highly similar fold (*Del Mármol et al., 2021*). As Orco shows higher sequence similarity to Grs – the ancestral family of insect chemosensory receptors from which Ors derived (*Brand et al., 2018*; *Dunipace et al., 2001*; *Robertson et al., 2003*) – we used *A. bakeri* Orco as the query structure in our analysis.

In our previous work (*Benton et al., 2020*), we generated ab initio protein models of Orco and candidate homologs in various unicellular eukaryotes using trRosetta (*Yang et al., 2020*) and RaptorX (*Källberg et al., 2012*). We therefore first examined the AlphaFold2 structural model of *A. bakeri* Orco (*Figure 1C*; *Jumper et al., 2021*; *Varadi et al., 2022*). This model displays striking qualitative similarity to the experimental structure (PDB 6C70 chain A) (*Figure 1C*). We assessed structural similarity quantitatively using two algorithms: first, using pairwise structural alignment in Dali (*Holm, 2022*), we extracted the resultant Z-score (the sum of equivalent residue-wise $C_\alpha$-$C_\alpha$ distances between two proteins); second, we determined the template modeling (TM)-score from TM-align (*Zhang and Skolnick, 2004*; *Zhang and Skolnick, 2005*) (a measure of the global similarity of full-length proteins) (*Table 1*). These measures confirmed the visual impression that the modeled and experimental structures are almost identical (e.g., TM-score=0.96, where 1 would be a perfect match). We extended our

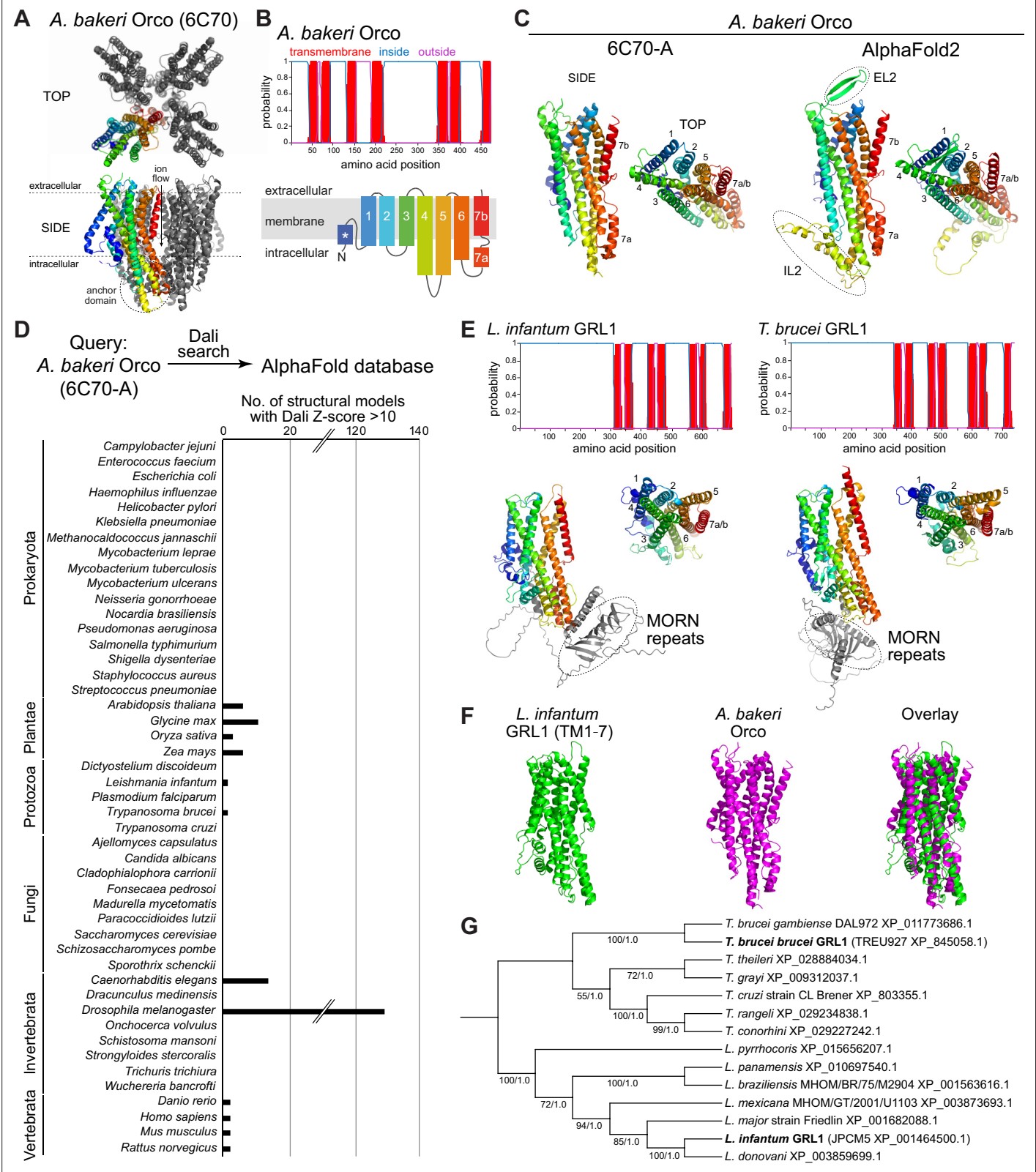

**Figure 1.** Structure-based screening for seven transmembrane domain ion channel (7TMIC) homologs. (**A**) Top view of a cryo-electronic microscopic (cryo-EM) structure of the homotetramer of Or co-receptor (Orco) from *A. bakeri* (derived from PDB 6C70; *Butterwick et al., 2018*), in which one subunit has a spectrum coloration (N-terminus [blue] to C-terminus [red]). The ion channel pore is formed at the interface of the four subunits. A side view is shown below. The anchor domain, comprising the cytoplasmic projections of TM4-6 and TM7a, forms most of the inter-subunit interactions in

*Figure 1 continued on next page*

*Figure 1 continued*

odorant receptors (Ors) (**Butterwick et al., 2018**; **Del Mármol et al., 2021**). (**B**) Top: output of transmembrane topology predictions of DeepTMHMM (**Hallgren et al., 2022**) for *A. bakeri* Orco. Bottom: schematic of the membrane topology of an Orco monomer, with the same spectrum coloration as in (**A**), reproduced from Figure 1a from **Benton et al., 2020**. Note that the seventh predicted helical region is divided into two in the cryo-EM structure: TM7a (located in the cytosol) and TM7b (located in the membrane). (**C**) Comparisons of side and top views of the cryo-EM structure of an *A. bakeri* Orco subunit (6C70-A) (left) and an AlphaFold2 protein structure prediction of *A. bakeri* Orco. Helical regions are numbered in the top views. Note the model contains the extracellular loop 2 (EL2) and intracellular loop 2 (IL2) regions that were not able to be accurately visualized in the cryo-EM structure (**Butterwick et al., 2018**). Quantitative comparisons of structures are provided in *Table 1*. (**D**) Summary of the results of the screen for Orco/Or-like protein folds in the AlphaFold Protein Structure Database for the indicated species using Dali (**Holm, 2022**). The threshold of Dali Z-score >10 was informed by inspection of the results of the screen (see Results). Raw outputs of the screen are provided in *Source data 2*. (**E**) Top: transmembrane topology predictions of the single screen hits from the *Trypanosoma* species *Leishmania infantum* and *Trypanosoma brucei brucei*. Bottom: AlphaFold2 structural models of these proteins, displayed as in (**C**). The long N-terminal region contains tandem Membrane Occupation and Recognition Nexus (MORN) repeats and sequence of unknown structure (gray); these are masked in the top view of the models. (**F**) Visual comparison of the *L. infantum* GRL1 AlphaFold2 model (the N-terminal region is masked) with the *A. bakeri* Orco structure, aligned with Coot (**Emsley et al., 2010**). Quantitative comparisons of structures are provided in *Table 1*. (**G**) Consensus phylogeny of putative trypanosome homologs. The primary sequence database was assembled using *L. infantum* GRL1 (XP_001464500.1) and *T. brucei brucei* GRL1 (XP_845058.1) as query sequences (highlighted in bold). Branch support values refer to maximum likelihood UFboot/Bayesian posterior probabilities. Note that although the *Trypanosoma cruzi* homolog (XP_803355.1) was not identified in the original Dali screen, visual inspection of the corresponding AlphaFold2 model (A0A2V2WL40) revealed the same global fold.

The online version of this article includes the following source data for figure 1:

**Source data 1.** FASTA file containing the amino acid sequences for validated trypanosome GRLs used in phylogenetic analyses.

**Source data 2.** FASTA file containing the multiple sequence alignment of trypanosome GRLs.

**Source data 3.** Newick tree file containing the maximum likelihood phylogeny of trypanosome GRLs.

**Source data 4.** NEXUS tree file containing the Bayesian phylogeny of trypanosome GRLs.

---

assessment of available (or newly generated) AlphaFold2 models to other well-established members of the 7TMIC family from animals as well as much more divergent unicellular 7TMIC homologs previously identified (**Benton et al., 2020**; *Source data 1*). Using the same quantitative assessments, these all displayed substantial tertiary structural similarity to *A. bakeri* Orco (*Table 1*), reinforcing our previous conclusions that these proteins form part of the same superfamily. Moreover, the observation that multiple distinct algorithms (AlphaFold2, trRosetta, and RaptorX) predict the same global fold of these proteins strengthens confidence in the validity of ab initio structural models.

We proceeded to screen the AlphaFold Protein Structure Database for other proteins that are structurally similar to *A. bakeri* Orco using the hierarchical search function in Dali (**Holm, 2022**). This algorithm currently permits pairwise alignment of Orco to the complete predicted structural proteomes of 47 species – encompassing several vertebrates, invertebrates, plants, unicellular eukaryotes and prokaryotes – returning hits ordered by Z-score (*Source data 2*). We focused on those hits with a Z-score of >10 (*Figure 1D*). This threshold successfully captured known 7TMICs, while removing a large number of proteins (generally with a much lower Z-score) that did not fulfill other criteria for structural similarity, as described below. Of the expected hits, within the *D. melanogaster* structural proteome we recovered all models of the members of the Or and Gr repertoires. From *Caenorhabditis elegans*, we found all members of the gustatory receptor (GUR) family (**Robertson et al., 2003**) – including the photoreceptor LITE-1 (formerly GUR-2) (**Edwards et al., 2008**; **Gong et al., 2016**; **Liu et al., 2010**) – and the serpentine receptor R (SRR) family (which are of unknown function, but display diverse neuronal and non-neuronal expression patterns **Vidal et al., 2018**; *Figure 1D* and *Source data 2*). From the four plant species screened, all members of the DUF3537 family were successfully identified (*Figure 1* and *Source data 2*). Inspection of several models below our Z-score threshold indicated that the proteins (typically multipass membrane proteins) have likely spurious resemblance to subregions of Orco rather than displaying similarity in their overall fold.

As will be illustrated below for individual novel candidate 7TMIC homologs, other hits were subsequently analyzed for their fulfillment of several criteria: (i) the presence of seven predicted TM domains, (ii) a predicted intracellular location of the N-terminus, and (iii) longer intracellular than extracellular loops (like insect Ors [**Otaki and Yamamoto, 2003**], while also recognizing that intracellular loops can vary enormously in length in homologs [**Benton et al., 2020**]). For hits that fulfilled these criteria, 'reverse' searching of the *D. melanogaster* structural proteome with Dali was performed to verify that Ors and Grs were structurally the most similar proteins in this species (*Source data 3*). We

**Table 1.** Quantitative structural comparisons of candidate seven transmembrane domain ion channel (7TMIC) homologs. Summary of amino acid identity (%), Dali Z-score, and TM-align TM-score of the indicated experimentally determined or ab initio-predicted structures of 7TMIC homologs (or negative-control, unrelated proteins) compared to *A. bakeri* Or co-receptor (Orco). The Orco cryo-electronic microscopic (cryo-EM) structure chain A (6C70-A) (*Butterwick et al., 2018*) was used as the query in all comparisons. Protein models are provided in *Source data 1*. Note the nomenclature of unicellular eukaryotic 7TMICs is tentative; identical names (e.g., GRL1) do not imply orthology. Typically, a Z-score >20 indicates that the two proteins being compared are definitely homologous, 8–20 that they are probably homologous, and 2–8 is a 'gray area' influenced by protein size and fold (*Holm, 2020*). TM-scores of 0.5–1 indicate that the two proteins being compared adopt generally the same fold, while TM-scores of 0–0.3 indicate random structural similarity (*Zhang and Skolnick, 2004*; *Zhang and Skolnick, 2005*). For the negative controls, the amino acid identity differs slightly between the experimentally determined and ab initio-predicted proteins because of small differences in sequence coverage.

| Category | Protein | Model or PDB | Method or algorithm | Comparison to *A. bakeri* Orco (6C70-A) | | |
|---|---|---|---|---|---|---|
| | | | | Amino acid identity (%) | Dali Z-score | TM-align TM-score |
| | *A. bakeri* Orco | 61b81_unrelaxed_rank_1_model_2 | AlphaFold2 | 100 | 50.7 | 0.96 |
| | *M. hrabei* Or5 | 7LIC-A | Cryo-EM | 19 | 36.3 | 0.81 |
| Positive controls (known 7TMIC) | *Drosophila melanogaster* Gr64a | AF-P83293-F1-model_v4 | AlphaFold2 | 13 | 29.6 | 0.79 |
| | *N. vectensis* GRL1 | AF-A7S7G0-F1-model_v4 | AlphaFold2 | 10 | 31.3 | 0.78 |
| Unicellular eukaryotic 7TMIC | *Thecamonas trahens* GRL1 | AF-A0A0L0DUY0-F1-model_v3 | AlphaFold2 | 9 | 23.2 | 0.71 |
| | *T. trahens* GRL2 | AF-A0A0L0DQC1-F1-model_v3 | AlphaFold2 | 12 | 25.3 | 0.70 |
| | *T. trahens* GRL3 | AF-A0A0L0D5B5-F1-model_v3 | AlphaFold2 | 14 | 13.1 | 0.50 |
| | *T. trahens* GRL4 | AF-A0A0L0D5H0-F1-model_v3 | AlphaFold2 | 9 | 9.9 | 0.53 |
| | *T. trahens* GRL5 | AF-A0A0L0DD38-F1-model_v3 | AlphaFold2 | 10 | 12.2 | 0.56 |
| | *T. trahens* GRL6 | AF-A0A0L0DJ52-F1-model_v3 | AlphaFold2 | 8 | 15.6 | 0.57 |
| | *V. brassicaformis* GRL1 | AF-A0A0G4FIT4-F1-model_v3 | AlphaFold2 | 10 | 9.1 | 0.47 |
| | *V. brassicaformis* GRL2 | AF-A0A0G4ECU2-F1-model_v3 | AlphaFold2 | 11 | 14.4 | 0.57 |
| | *V. brassicaformis* GRL3 | AF-A0A0G4FWI7-F1-model_v3 | AlphaFold2 | 14 | 23.8 | 0.74 |
| | *V. brassicaformis* GRL4 | AF-A0A0G4EU86-F1-model_v3 | AlphaFold2 | 10 | 18.5 | 0.70 |
| | *V. brassicaformis* GRL5 | AF-A0A0G4FBY6-F1-model_v3 | AlphaFold2 | 10 | 18.5 | 0.68 |
| | *V. brassicaformis* GRL6 | AF-A0A0G4G8W6-F1-model_v3 | AlphaFold2 | 8 | 21.4 | 0.70 |
| | *Micromonas pusilla* GRL1 | AF-C1MGH9-F1-model_v3 | AlphaFold2 | 12 | 11.3 | 0.60 |
| | *Chloropicon primus* GRL1 | AF-A0A5B8MFA4-F1-model_v3 | AlphaFold2 | 10 | 18.1 | 0.71 |
| | *L. infantum* GRL1 | AF-A4HWQ9-F1-model_v3 | AlphaFold2 | 6 | 13.5 | 0.64 |
| | *T. brucei* GRL1 | AF-Q57U78-F1-model_v3 | AlphaFold2 | 9 | 13.4 | 0.62 |

*Table 1 continued on next page*

Table 1 continued

| Category | Protein | Model or PDB | Method or algorithm | Comparison to *A. bakeri* Orco (6C70-A) | | |
|---|---|---|---|---|---|---|
| | | | | Amino acid identity (%) | Dali Z-score | TM-align TM-score |
| | *D. melanogaster* Grl36a | AF-Q8INZ1-F1-model_v3 | AlphaFold2 | 9 | 19.5 | 0.67 |
| | *D. melanogaster* Grl36b | AF-Q8INY2-F1-model_v3 | AlphaFold2 | 8 | 15.2 | 0.62 |
| | *D. melanogaster* Grl40a | AF-Q0E8M7-F1-model_v3 | AlphaFold2 | 8 | 19.5 | 0.66 |
| | *D. melanogaster* Grl43a | AF-Q9V4Q0-F1-model_v3 | AlphaFold2 | 10 | 19.9 | 0.69 |
| | *D. melanogaster* Grl58a | AF-Q9W2A4-F1-model_v3 | AlphaFold2 | 8 | 15.0 | 0.60 |
| | *D. melanogaster* Grl62a | AF-B7Z0I0-F1-model_v3 | AlphaFold2 | 8 | 19.4 | 0.69 |
| | *D. melanogaster* Grl62b | AF-B7Z0I1-F1-model_v3 | AlphaFold2 | 11 | 19.1 | 0.66 |
| | *D. melanogaster* Grl62c | AF-Q6ILZ2-F1-model_v3 | AlphaFold2 | 10 | 17.2 | 0.63 |
| | *D. melanogaster* Grl65a | AF-Q8IQ72-F1-model_v3 | AlphaFold2 | 11 | 25.9 | 0.74 |
| Fly Grl | *D. melanogaster* GrlHz | AF-Q9W1W8-F1-model_v3 | AlphaFold2 | 7 | 22.5 | 0.74 |
| | *Homo sapiens* PHTF1 | AF-Q9UMS5-F1-model_v3 | AlphaFold2 | 7 | 12.9 | 0.63 |
| | *H. sapiens* PHTF2 | AF-Q8N3S3-F1-model_v3 | AlphaFold2 | 8 | 12.0 | 0.62 |
| PHTF | *D. melanogaster* Phtf | AF-Q9V9A8-F1-model_v3 | AlphaFold2 | 5 | 11.8 | 0.63 |
| | *Bos taurus* Rhodopsin | 1F88-A | X-ray crystal | 9 | 2.1 | 0.31 |
| | | AF-P02699-F1-model_v4 | AlphaFold2 | 9 | <2.0 | 0.19 |
| | *Chlamydomonas reinhardtii* ChR2 | 6EID-A | X-ray crystal | 7 | 3.6 | 0.27 |
| | | AF-Q8RUT8-F1-model_v4 | AlphaFold2 | 9 | 3.4 | 0.10 |
| | *H. sapiens* Frizzled4 | 6BD4 | X-ray crystal | 8 | 4.0 | 0.34 |
| | | AF-Q9ULV1-F1-model_v4 | AlphaFold2 | 5 | 2.9 | 0.19 |
| | *H. sapiens* AdipR | 5LXG | X-ray crystal | 2 | 3.6 | 0.29 |
| | | AF-Q96A54-F1-model_v4 | AlphaFold2 | 2 | <2.0 | 0.14 |
| | *Escherichia coli* GlpG | 2XOV | X-ray crystal | 5 | 3.5 | 0.27 |
| | | AF-P09391-F1-model_v4 | AlphaFold2 | 6 | 3.3 | 0.13 |
| | *Mus musculus* TRPV3 | 6LGP-D | Cryo-EM | 10 | 2.7 | 0.27 |
| | | AF-Q8K424-F1-model_v4 | AlphaFold2 | 14 | 2.3 | 0.08 |
| | *M. musculus* Piezo | 6BPZ-B | Cryo-EM | 5 | 4.0 | 0.27 |
| | | AF-E2JF22-F1-model_v4 | AlphaFold2 | 5 | 2.3 | 0.08 |
| Negative controls (non-7TMIC) | *B. taurus* CNGA/CNGB | 7O4H-A | Cryo-EM | 9 | 2.8 | 0.24 |
| | | AF-Q00194-F1-model_v4 | AlphaFold2 | 9 | 3.3 | 0.11 |

next qualitatively assessed the predicted tertiary structural similarity to *A. bakeri* Orco (*Figure 1A–C*; *Butterwick et al., 2018*), verifying: (i) the characteristic packing of the TMs, (ii) the projection of the long TM4, TM5, and TM6 below the main bundle of helices (forming the 'anchor' domain where most inter-subunit contacts occur in complexes; *Butterwick et al., 2018*; *Del Mármol et al., 2021*), and (iii) the exceptional splitting of TM7 into two subregions (TM7a, part of the anchor domain, and TM7b, which lines the ion conduction pathway; *Butterwick et al., 2018*; *Del Mármol et al., 2021*). Structures were also quantitatively compared to *A. bakeri* Orco, as described above (*Table 1*). As negative controls, we also performed comparisons with a variety of other multipass membrane

proteins belonging to other superfamilies, including several with seven TMs (e.g., Rhodopsin, Frizzled, and the Adiponectin receptor) (*Table 1*). The new candidate homologs all displayed quantitative measures of similarity that were within the range of previously identified 7TMIC homologs, and clearly superior to the scores of negative control proteins (*Table 1*). We now present these candidate homologs from different species and the potential evolutionary and biological implications for the 7TMIC family, bearing in mind the caveat that some of these may represent cases of structural convergence (discussed below).

Extending our previous discovery of 7TMICs in various single-celled eukaryotes (informally grouped here under the term Protozoa) (*Benton et al., 2020*), we identified single proteins in two species belonging to the Trypanosomatida order: *L. infantum* and *T. brucei*, the causal agents in humans of trypanosomiasis (sleeping sickness) and visceral leishmaniasis (black fever), respectively (*Figure 1D–F* and *Table 1*). Beyond the 7TMIC-like protein fold (*Figure 1E–F* and *Table 1*), these proteins are characterized in their N-terminal regions by a Membrane Occupation and Recognition Nexus (MORN)-repeat domain, which is implicated in protein-protein interaction and possibly lipid binding (*Sajko et al., 2020*). BLAST searches identified homologous proteins only within trypanosomes (*Figure 1G*), consistent with our failure to recover these sequences in earlier primary structure-based screens for 7TMICs. We did not detect any structurally related proteins to Orco in Prokaryota or Fungi (previously, fungal GRLs were only identified in chytrids [*Benton et al., 2020*], which are not currently surveyed via Dali). Together, these results reinforce our previous conclusion (*Benton et al., 2020*) that 7TMICs evolved in or prior to the last eukaryotic common ancestor, and provide a first example of fusion of this TM protein fold with a distinct, cytoplasmic protein domain.

## PHTF proteins are candidate vertebrate 7TMICs

Given previous lack of success in identifying homologs of 7TMICs within any chordate genome, we were intrigued that our screen recovered two hits from *H. sapiens* (and orthologous proteins of the three other vertebrate species screened) (*Figure 1D* and *Source data 1–3*). The human proteins, PHTF1 and PHTF2, are very similar to each other (54.1% amino acid identity) and have the characteristic topology of 7TMICs (*Figure 2A*). The next most similar vertebrate proteins to Orco had substantially lower Dali Z-scores than PHTFs and represented a variety of likely spurious matches (*Source data 2*). The single *D. melanogaster* ortholog (Phtf) (*Manuel et al., 2000*) displays a similar topology to the vertebrate proteins (*Figure 2A*), and is the next most similar protein model to *A. bakeri* Orco after the *D. melanogaster* Grs, Ors, and Grls (see next section) (*Source data 2*). PHTF is an acronym of 'Putative Homeodomain Transcription Factor', a name originally proposed because of presumably artifactual sequence similarity of a short region around TM4 to homeodomain DNA-binding sequences (*Raich et al., 1999*); subsequent histological and biochemical studies (discussed below) established that PHTF1 is an integral membrane protein (*Oyhenart et al., 2003*).

To visually compare AlphaFold2 models of PHTF orthologs with *A. bakeri* Orco, we masked the long (>300 amino acid) first intracellular loop (*Figure 2A*), whose structure is mostly unpredicted but contains a few α-helical regions, as well as the ~100-residue N-terminus (*Figure 2B*). This visualization revealed the clear similarity in the organization of the seven TM helical core of the protein, including the split TM7 (*Figure 2B–C*), which was verified by quantitative structural comparisons (*Table 1*).

In contrast to other, taxon-restricted members of the 7TMIC superfamily, highly conserved PHTF homologs were found across Eukaryota, including in Bilateria, Cnidaria, and several unicellular species (*Figure 2D*). Phylogenetic analyses of a representative PHTF protein sequence dataset revealed that there is a single eukaryotic PHTF clade (*Figure 2E* and *Figure 2—figure supplements 1–3*). Bayesian and maximum likelihood phylogenetics largely agree on the topology of this tree and suggest that the PHTF1-PHTF2 duplication occurred specifically in the jawed vertebrate lineage (Gnathostomata) (*Figure 2E*).

Previous tissue-specific RNA expression analysis by northern blotting of *H. sapiens PHTF1* and *PHTF2* revealed enrichment in testis and muscle, respectively (*Manuel et al., 2000*). We confirmed and extended these conclusions by analyzing publicly available bulk RNA-sequencing (RNA-seq) datasets: *PHFT1* is most abundantly detected in cerebellum and testis, and *PHTF2* in skeletal muscle and arteries (*Figure 2F* and *Figure 2—figure supplement 4*). *D. melanogaster Phtf* displays highly enriched expression in the testis, and much lower expression in neural tissues in the FlyAtlas 2.0 bulk RNA-seq datasets (*Figure 2F* and *Figure 2—figure supplement 5*; *Krause et al., 2022*), potentially

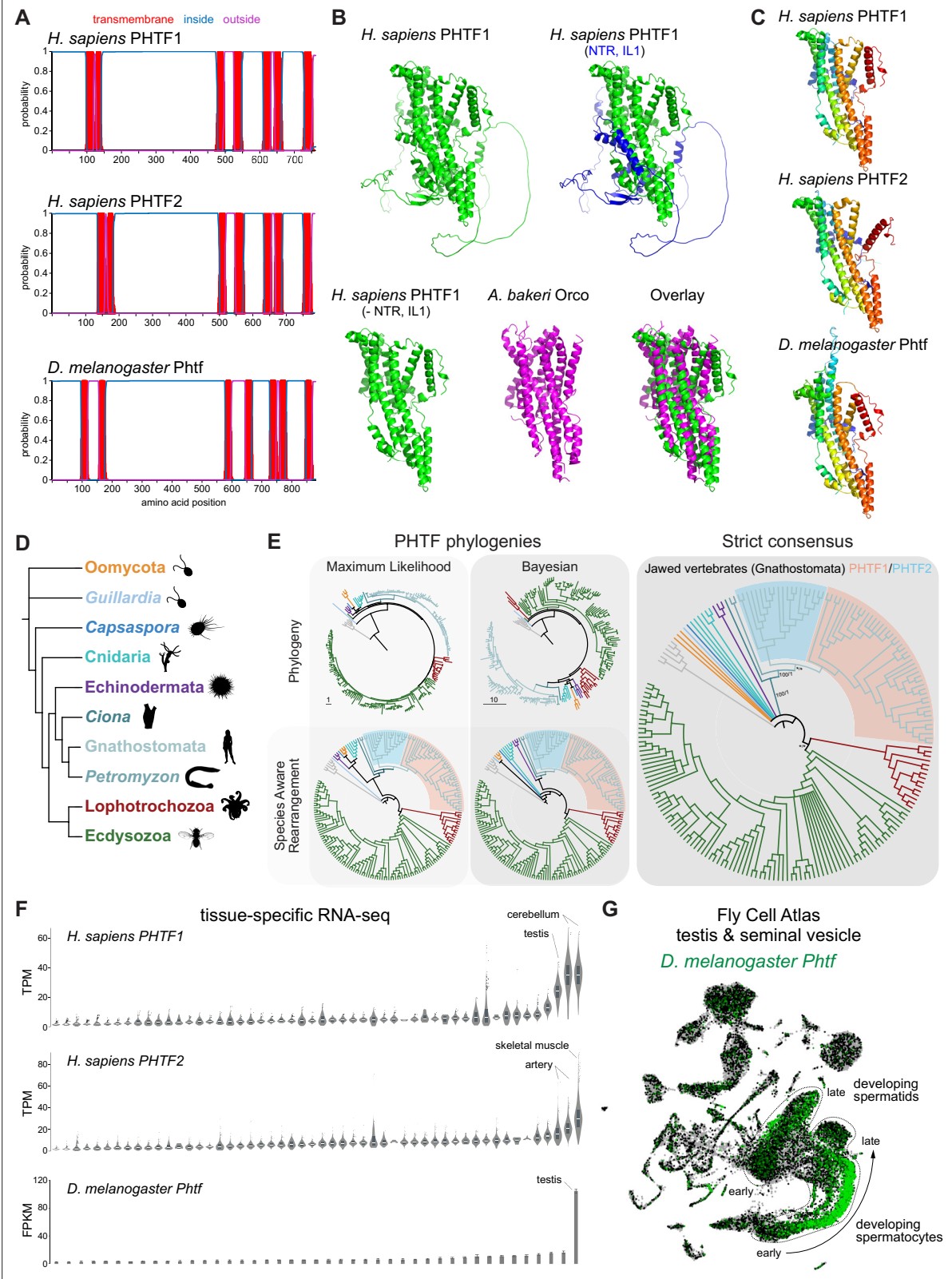

**Figure 2.** PHTF proteins are candidate vertebrate seven transmembrane domain ion channels (7TMICs). (**A**) DeepTMHMM-predicted transmembrane topology of PHTF proteins. (**B**) Top: AlphaFold2 predicted structure of *H. sapiens* PHTF1; in the image on the right the long N-terminal region (NTR) and intracellular loop 1 (IL1) are highlighted in blue; these sequences contain a few predicted helical regions but are of largely unknown structure. Bottom: visual comparison of the *H. sapiens* PHTF1 AlphaFold2 structure (in which the NTR and IL1 are masked) with the *A. bakeri* Or co-receptor (Orco)

*Figure 2 continued on next page*

*Figure 2 continued*

structure. (**C**) AlphaFold2 structures of PHTF proteins in which the NTR and IL1 are masked. Quantitative comparisons of these structures to the cryo-electronic microscopic (cryo-EM) Orco structure are provided in *Table 1*. (**D**) Major taxa/species in which a PHTF homolog was identified (see sequence databases in *Figure 2—source data 1*). Silhouette images in this and other figures are from Phylopic (https://www.phylopic.org/). (**E**) Phylogenies of a representative set of PHTF sequences. The sequence database was constructed using the *D. melanogaster* and *H. sapiens* PHTF query sequences. Top left: maximum likelihood phylogeny (JTT + R10 model) and Bayesian phylogeny. The scale bars represent the average number of substitutions per site. Bottom left: phylogenies where weakly supported branches (<95/0.95) have been rearranged and polytomies resolved in a species tree-aware manner. Right: strict consensus of the species tree-aware phylogenies. There is a single eukaryotic PHTF clade and the PHTF1-2 split occurred in the jawed vertebrate lineage. However, this interpretation relies on the rearrangement of the weakly supported jawless vertebrate PHTF branch. Therefore, an alternative but weakly supported hypothesis is that the duplication occurred in a common vertebrate ancestor and a single PHTF copy was lost in jawless vertebrates. Select branch support values are present on key branches and refer to maximum likelihood UFboot/Bayesian posterior probabilities. Asterisks indicate that branch support was below the threshold for species-aware rearrangement. The fully annotated trees are available in *Figure 2—figure supplements 1–3*. (**F**) Summary of tissue-enriched RNA expression of *H. sapiens PHTF1* and *PHTF2* (data are from the GTex Portal; the fully annotated dataset is provided in *Figure 2—figure supplement 4*) and *D. melanogaster Phtf* (data from the Fly Atlas 2.0; the fully annotated dataset is provided in *Figure 2—figure supplement 5*). (**G**) Left: Uniform Manifold Approximation and Projection (UMAP) representation of RNA-seq datasets from individual cells of the *D. melanogaster* testis and seminal vesicle generated as part of the Fly Cell Atlas (10× relaxed dataset) (*Li et al., 2022*) colored for expression of *Phtf*. Simplified annotations of cell clusters displaying the highest levels of *Phtf* expression are adapted from *Li et al., 2022*; unlabeled clusters represent non-germline cell types of the testis.

The online version of this article includes the following source data and figure supplement(s) for figure 2:

**Source data 1.** FASTA file containing the amino acid sequences of validated eukaryotic PHTFs.

**Source data 2.** FASTA file containing the representative amino acid sequences of eukaryotic PHTFs used in phylogenetic analyses.

**Source data 3.** FASTA file containing the multiple sequence alignment of eukaryotic PHTFs.

**Source data 4.** Newick tree file containing the maximum likelihood phylogeny of eukaryotic PHTFs.

**Source data 5.** NOTUNG tree file containing the species-aware phylogeny of eukaryotic PHTFs, based on the maximum likelihood phylogeny.

**Source data 6.** NEXUS tree file containing the Bayesian phylogeny of eukaryotic PHTFs.

**Source data 7.** NOTUNG tree file containing the species-aware phylogeny of eukaryotic PHTFs, based on the Bayesian phylogeny.

**Source data 8.** Newick tree file containing the strict consensus of the species-aware phylogenies of eukaryotic PHTFs.

**Figure supplement 1.** Fully annotated phylogenetic trees for PHTF homologs.

**Figure supplement 2.** Fully annotated species-aware trees for PHTF homologs.

**Figure supplement 3.** Strict consensus of the species-aware trees for PHTF homologs.

**Figure supplement 4.** Tissue-specific RNA expression of *H. sapiens PHTF1* and *PHTF2*.

**Figure supplement 5.** Tissue-specific RNA expression of *D. melanogaster Phtf* and *Grls*.

indicating a closer functional relationship to *PHTF1* than *PHTF2*. Higher resolution expression analysis of *Phtf* in male reproductive tissue, using the Fly Cell Atlas (*Li et al., 2022*), revealed the most prominent expression in developing spermatocytes and spermatids (*Figure 2G*). The transcript expression of *D. melanogaster Phtf* is concordant with detection of rat (*Rattus norvegicus*) PHTF1 protein from primary spermatocytes to the end of spermatogenesis, predominantly localized to the endoplasmic reticulum (*Oyhenart et al., 2005b*; *Oyhenart et al., 2003*). The N-terminal region of mouse (*M. musculus*) PHTF1 associates with the testis-enriched FEM1B E3 ubiquitin ligase and is suggested to recruit it to the endoplasmic reticulum (*Oyhenart et al., 2005a*). Overexpression and/or knock-down studies of PHTF1 and PHTF2 in cell lines hint at roles in regulating cell proliferation and survival, and possible links to various cancers (*Chi et al., 2020*; *Huang et al., 2015*). However, the biological function of any PHTF protein in any organism is unclear. Nevertheless, PHTFs represent the first candidate homologs of insect Ors/Grs in chordates, indicating that they might not have been completely lost from this lineage, as previously thought (*Benton, 2015*; *Robertson, 2015*); we suggest they also act as ion channels.

## Novel sets of candidate insect chemoreceptors

Within the hits of our screen of *D. melanogaster* protein structures, we noticed 10 proteins that do not belong to the canonical Gr or Or families (*Source data 1–3*). These proteins have a similar length and TM topology as Grs and Ors (*Figure 3A*). Visual inspection and quantitative analyses confirmed that their predicted fold is very similar to that of *A. bakeri* Orco (*Figure 3A* and *Table 1*). As they almost completely lack other defining sequence features of these families (see below), we named these

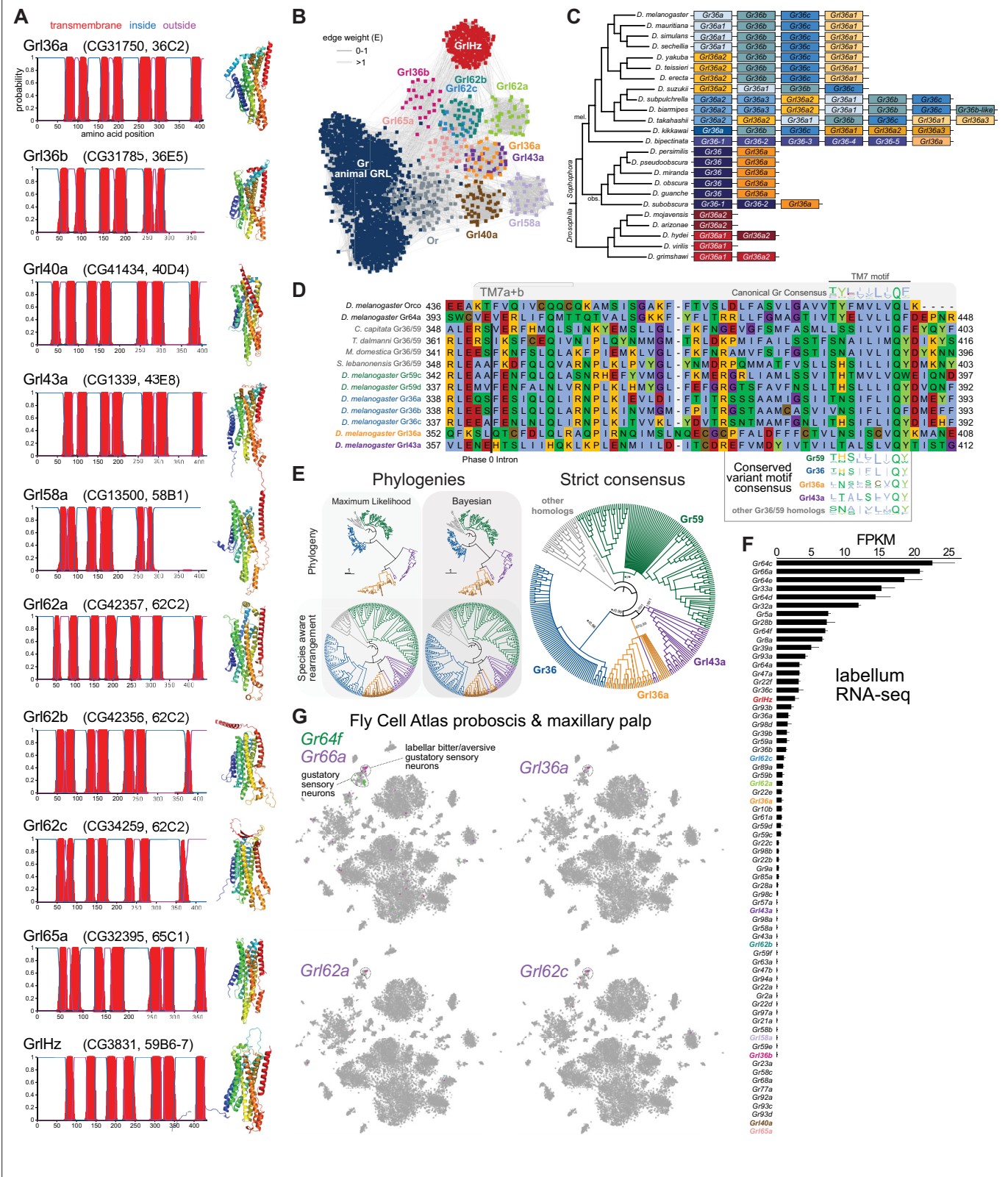

**Figure 3.** Insect Grls are highly divergent, candidate chemosensory receptors. (**A**) Proposed nomenclature of *D. melanogaster* Grls (the original gene name and cytological location are in parentheses), with corresponding DeepTMHMM-predicted transmembrane topologies and AlphaFold2 structural models. Note that TM7 is not predicted for Grl36b and Grl58a by DeepTMHMM, but is predicted – with the characteristic TM7a/7b split – in the structural model (as well as predicted by Phobius [data not shown]). Quantitative comparisons of these structures to the cryo-electronic microscopic

*Figure 3 continued on next page*

*Figure 3 continued*

(cryo-EM) Or co-receptor (Orco) structure are provided in *Table 1*. (**B**) Sequence similarity network of Grls, Grs, and Ors (including Orco). The network was generated using an all-to-all comparison made by MMSeqs2 as implemented by gs2. The connections represent E-values where the weakest connections (arbitrarily defined as edge weights >1) are colored in lighter gray. Lack of connection between two nodes indicates that those two sequences could not be identified as having any significant sequence similarity under the most sensitive MMSeqs2 settings. Nodes and edges are arranged in a prefuse force-directed layout. The graph splitting tree is visualized in *Figure 3—figure supplement 5*; however, we do not place high confidence in the phylogenetic accuracy of the tree due to the likely effects of long branch attraction. The evolution of GrlHolozoa (GrlHz) is described in *Figure 3—figure supplement 1*, with detailed phylogenies in *Figure 3—figure supplements 2–4*. (**C**) Schematic of the gene arrangement of *Grl36a* and *Gr36* homologs in drosophilids. Color coding reflects relatedness with respect to major speciation and gene duplication events; colors match the phylogenetic tree branches in *Figure 3—figure supplement 6B–C*. The *Drosophila* subgenus entirely lacks Gr36 homologs (see *Figure 3—figure supplement 6*). (**D**) Alignment of the C-terminal region of *D. melanogaster* Orco, Gr64a, select insect Gr36/Gr59 homologs, and *D. melanogaster* Grl36a and Grl43a, extracted from a larger alignment available in *Figure 3—source data 5*. The black bar shows the common location of a phase 0 intron, which is presumably homologous in different sequences. The canonical TM7 motif of the Gr family (represented as relative amino acid frequencies extracted from WebLogo) is shown above the sequence, and the variant motifs of different Gr or Grl ortholog groups are shown below. (**E**) Phylogenies of Gr36, Gr59c/d, Grl36a, Grl43a and homologous non-drosophilid sequences (color-coded as in (**D**)). The sequence database was assembled using *D. melanogaster* Gr36a, Grl36a, and Grl43a as the query sequences. Top left: maximum likelihood phylogeny (JTT + F + R7 model) and Bayesian phylogeny. The scale bars represent average number of substitutions per site. Bottom left: phylogenies where weakly supported branches (<95/0.95) have been rearranged and polytomies resolved in a species tree-aware manner. Right: strict consensus of the species tree-aware phylogenies. These analyses support that Gr36 and Grl36a/43a are sister clades, which likely split after Gr59c/d diverged from the ancestral lineage. Sequences are colored as in (**D**). Select branch support values are present on key branches and refer to maximum likelihood UFboot and Bayesian posterior probabilities, in this order. Asterisks indicate that branch support was below the threshold for species-aware rearrangement. A simplified schematic of gene duplication and loss is illustrated in *Figure 3—figure supplement 6F*. The fully annotated trees are available in *Figure 3—figure supplements 7–9*. (**F**) Histogram of *Gr* and *Grl* expression levels in adult proboscis and maxillary palps determined by bulk RNA-sequencing (RNA-seq). Mean values ± SD of fragments per kilobase of transcript per million mapped reads (FPKM) are plotted; n=3 biological replicates. Data is from *Dweck et al., 2021*. (**G**) Left: t-distributed stochastic neighbor embedding (tSNE) representation of RNA-seq datasets from individual cells of the *D. melanogaster* proboscis and maxillary palp – generated as part of the Fly Cell Atlas (10× stringent dataset) (*Li et al., 2022*) – colored for expression of the indicated genes. *Gr64f* and *Gr66a* are broad markers of 'sweet/appetitive' and 'bitter/aversive' gustatory sensory neurons, respectively. Transcripts for three *Grl*s are detected in subsets of bitter/aversive neurons. Annotations of cell clusters are adapted from *Li et al., 2022*; unlabeled clusters represent other non-gustatory sensory neuron or non-neuronal cell types of this tissue.

The online version of this article includes the following source data and figure supplement(s) for figure 3:

**Source data 1.** FASTA file containing the amino acid sequences used in the network and graph splitting analysis of gustatory receptors (Grs), odorant receptors (Ors), and Grls.

**Source data 2.** Tab delimited text file containing the sequence similarity network of gustatory receptors (Grs), odorant receptors (Ors), and Grls.

**Source data 3.** Tab delimited text file containing the annotation for the sequence similarity network of gustatory receptors (Grs), odorant receptors (Ors), and Grls.

**Source data 4.** Newick tree file containing the graph splitting tree of odorant receptors (Ors), gustatory receptors (Grs), and Grls, derived from the sequence similarity network by gs2.

**Source data 5.** FASTA file containing the multiple sequence alignment used for illustrating intron and transmembrane domain 7 (TM7) motif conservation between gustatory receptors (Grs) and Grls.

**Source data 6.** FASTA file containing the amino acid sequences of Gr36, Gr59, Grl36a, and Grl43a homologs.

**Source data 7.** FASTA file containing the multiple sequence alignment of Gr36, Gr59, Grl36a, and Grl43a homologs.

**Source data 8.** Newick tree file containing the maximum likelihood phylogeny of Gr36, Gr59, Grl36a, and Grl43a homologs.

**Source data 9.** NOTUNG tree file containing the species-aware phylogeny of Gr36, Gr59, Grl36a, and Grl43a homologs, based on the maximum likelihood phylogeny.

**Source data 10.** NEXUS tree file containing the Bayesian phylogeny of Gr36, Gr59, Grl36a, and Grl43a homologs.

**Source data 11.** NOTUNG tree file containing the species-aware phylogeny of Gr36, Gr59, Grl36a, and Grl43a homologs, based on the Bayesian phylogeny.

**Source data 12.** Newick tree file containing the strict consensus of the species-aware phylogenies of Gr36, Gr59, Grl36a, and Grl43a homologs.

**Figure supplement 1.** Evolution of GrlHolozoa (GrlHz), a family of Grl seven transmembrane domain ion channel (7TMIC) not restricted to flies.

**Figure supplement 1—source data 1.** FASTA file containing the amino acid sequences of validated holozoan GrlHolozoa (GrlHz).

**Figure supplement 1—source data 2.** FASTA file containing the representative amino acid sequences of holozoan GrlHolozoa (GrlHz) used in phylogenetic analyses.

**Figure supplement 1—source data 3.** FASTA file containing the multiple sequence alignment of holozoan GrlHolozoa (GrlHz).

**Figure supplement 1—source data 4.** Newick tree file containing the maximum likelihood phylogeny of holozoan GrlHolozoa (GrlHz).

**Figure supplement 1—source data 5.** NOTUNG tree file containing the species-aware phylogeny of holozoan GrlHolozoa (GrlHz), based on the maximum likelihood phylogeny.

*Figure 3 continued on next page*

*Figure 3 continued*

**Figure supplement 1—source data 6.** NEXUS tree file containing the Bayesian phylogeny of holozoan GrlHolozoa (GrlHz).

**Figure supplement 1—source data 7.** NOTUNG tree file containing the species-aware phylogeny of holozoan GrlHolozoa (GrlHz), based on the Bayesian phylogeny.

**Figure supplement 1—source data 8.** Newick tree file containing the strict consensus of the species-aware phylogenies of holozoan GrlHolozoa (GrlHz).

**Figure supplement 2.** Fully annotated phylogenetic trees for GrlHolozoa (GrlHz) homologs.

**Figure supplement 3.** Fully annotated species-aware trees for GrlHolozoa (GrlHz) homologs.

**Figure supplement 4.** Strict consensus of the species-aware trees for GrlHolozoa (GrlHz) homologs.

**Figure supplement 5.** Fully annotated graph splitting tree for odorant receptors (Ors), gustatory receptors (Grs), and Grls.

**Figure supplement 6.** The evolution of Gr36, Gr59, Grl36a, and Grl43a.

**Figure supplement 6—source data 1.** FASTA file containing the amino acid sequences of Gr36 homologs.

**Figure supplement 6—source data 2.** FASTA file containing the multiple sequence alignment of Gr36 homologs.

**Figure supplement 6—source data 3.** Newick tree file containing the maximum likelihood phylogeny of Gr36 homologs.

**Figure supplement 6—source data 4.** NOTUNG tree file containing the species-aware phylogeny of Gr36 homologs, based on the maximum likelihood phylogeny.

**Figure supplement 6—source data 5.** NEXUS tree file containing the Bayesian phylogeny of Gr36 homologs.

**Figure supplement 6—source data 6.** NOTUNG tree file containing the species-aware phylogeny of Gr36 homologs, based on the Bayesian phylogeny.

**Figure supplement 6—source data 7.** FASTA file containing the amino acid sequences of Grl36a homologs.

**Figure supplement 6—source data 8.** FASTA file containing the multiple sequence alignment of Grl36a homologs.

**Figure supplement 6—source data 9.** Newick tree file containing the maximum likelihood phylogeny of Grl36a homologs.

**Figure supplement 6—source data 10.** NOTUNG tree file containing the species-aware phylogeny of Grl36a homologs, based on the maximum likelihood phylogeny.

**Figure supplement 6—source data 11.** NEXUS tree file containing the Bayesian phylogeny of Grl36a homologs.

**Figure supplement 6—source data 12.** NOTUNG tree file containing the species-aware phylogeny of Grl36a homologs, based on the Bayesian phylogeny.

**Figure supplement 7.** Fully annotated phylogenetic trees for Gr36, Gr59, Grl36a, and Grl43a homologs.

**Figure supplement 8.** Fully annotated species-aware trees for Gr36, Gr59, Grl36a, and Grl43a homologs.

**Figure supplement 9.** Strict consensus of the species-aware trees for Gr36, Gr59, Grl36a, and Grl43a homologs.

---

Grl proteins, using the same cytogenetic-based gene nomenclature conventions of other chemosensory gene families (e.g., *Drosophila Odorant Receptor Nomenclature Committee, 2000*), with one exception (GrlHolozoa [GrlHz], see below).

For seven *D. melanogaster* Grls, BLAST searches identified homologs only in drosophilids; for two others (Grl40a and Grl65a) we recovered drosophilid and other fly homologs. By contrast, the Grl originally designated CG3831 has homologs across a wide range of Holozoa (i.e., animals and their closest single-celled, non-fungal relatives), including chordates (e.g., the lancelet *Branchiostoma floridae*) and single-cell eukaryotes (e.g., *Capsaspora owczarzaki*) (*Figure 3—figure supplements 1–4*), leading us to name it GrlHolozoa (GrlHz). A subset of GrlHz homologs bear a long N-terminal domain containing WD40 repeats, which form a structurally predicted beta-propeller domain that is typically involved in protein-protein interactions (*Figure 3—figure supplement 1D*; *Kim and Kim, 2020*).

Given that nine of these Grls are restricted to flies, a reasonable hypothesis is that they evolved from fly Grs. To infer their evolutionary origins, we therefore examined sequence similarity of Grls with a representative set of Grs, as well as Ors and other animal (i.e., non-insect) GRLs. We found that fly Grls share little or no obvious sequence similarity with any of these other 7TMICs, precluding confident standard phylogenetic analysis and leading us to use an all-to-all graph-based methodology, which does not require a multiple sequence alignment. This approach infers relationships between sequences based on pairwise sequence similarity. This analysis first generates an all-to-all sequence similarity network via MMseqs2, in which sequence families can be identified as clusters in a 2D projection (*Figure 3B*), and then a tree by recursive spectral clustering (*Figure 3—figure supplement 5*; see Methods; *Matsui and Iwasaki, 2020*; *Steinegger and Söding, 2017*). In the network,

we observed that several of the Grls were intermingled in clusters (e.g., Grl62b/Grl62c and Grl36a/Grl43a), suggesting relatively recent common ancestry (*Figure 3B*). These two clusters were recapitulated as clades in the graph splitting phylogeny (*Figure 3—figure supplement 5*). For Grl62a/b/c, the possibility of recent ancestry is consistent with the tandem genomic organization of the corresponding genes, which implies their evolution by gene duplication through non-allelic homologous recombination, similar to other families of chemosensory genes (*Nei et al., 2008*). None of the Grl clusters grouped with those of Ors, Grs, or other animal GRLs, rather connecting broadly, but weakly, with all other clusters (*Figure 3B*). Consistent with this clustering pattern, all Grls were placed near the presumed root of the graph splitting tree (*Figure 3—figure supplement 5*). This basal placement of Grls was inconsistent with their conservation only in flies, and is likely a phylogenetic artifact (see legend to *Figure 3—figure supplement 5*).

Although analysis of amino acid sequences did not provide evidence of ancestry between Grs and Grls, we noted that *Grl36a* was adjacent (separated by 306 bp) to the *Gr36a/b/c* cluster in the *D. melanogaster* genome. This proximity suggested that *Grl36a* might have arisen by gene duplication of a *Gr36*-like ancestor. Indeed, *Grl36a* homologs across drosophilid species were always found in tandem with *Gr36*-related genes in various arrangements (*Figure 3C*, *Figure 3—figure supplement 6*). To further investigate the hypothetical ancestry of *Grl36a* and *Gr36*, we first examined their gene structure. We incorporated into this analysis *Gr59c* and *Gr59d* homologs, which are closely related to *Gr36a/b/c* even though they are distantly located in the genome (*Robertson et al., 2003*), as well as *Grl43a*, the most closely related paralog to *Grl36a* (*Figure 3B*). The *Gr* family is characterized by the general, but not universal, conservation of three phase 0 introns near the 3' end of these genes (*Robertson et al., 2003*). *Gr36*, *Gr59c/d,* and homologous non-drosophilid genes possess only one of these introns, which corresponds to the second ancestral *Gr* intron located just before the exon encoding TM7. Both *D. melanogaster Grl36a* and *Grl43a* also have a phase 0 intron immediately before the TM7-encoding exon, which aligns with the *Gr* intron position on a multiple protein sequence alignment (*Figure 3D*), suggesting that these *Gr* and *Grl* introns are homologous. We next examined the TM7 motifs in these Grs and Grls. The canonical TM7 motif of Grs is TYhhhhhQF, where h is a hydrophobic residue (*Figure 3D*; *Robertson, 2019*; *Scott et al., 2001*). However, Gr36 and Gr59c/d share a variant motif, T(H/N)(S/A)hhhhQ(Y/F/W), and we observed a very similar motif in Grl36a and Grl43a (*Figure 3D*).

The genomic proximity of *Gr36* and *Grl36a*, and similarity in introns and TM7 motifs of these genes (as well as *Gr59c/d* and *Grl43a*) provide evidence that these genes have a relatively recent common ancestry within drosophilids. Phylogenetic analyses of this proposed clade support that a Grl36a/Grl43a clade is the sister clade to Gr36, and that this split occurred after the emergence of the Gr59c/d clade (*Figure 3E*, *Figure 3—figure supplements 6–9*). None of the other *Grl* genes are located adjacent to *Gr* genes, nor do the proteins possess a recognizable TM7 motif. Some other Grls might possess conserved introns of *Gr*s (e.g., *Grl40a* with the first ancestral intron, and *Grl36b* and *Grl65a* with the second ancestral intron [data not shown]), but we cannot conclude with confidence that these are homologous. Thus, the ancestry of most Grls remains unresolved. Nevertheless, the highly restricted taxonomic representation of nine of these Grls and their structural similarity to Grs support a model in which Grls have rapidly evolved and diverged from ancestral Grs.

To gain insight into the potential role(s) of Grls, we first examined their expression in tissue-specific bulk RNA-seq datasets from the FlyAtlas 2.0 (*Krause et al., 2022*). Most *Grl*s were expressed at very low (<1 fragment per kilobase of exon per million mapped fragments [FPKM]) or undetectable levels in essentially all tissues in these datasets, although *Grl36*b was detected in neuronal tissues (eye, brain, thoracicoabdominal ganglion) (*Figure 2—figure supplement 5*). The one exception was *GrlHz*, which was expressed (>8 FPKM) in various tissues (e.g., heart, ovary, testis, and larval fat body and garland cells [nephrocytes]). The unique expression and conservation properties of *GrlHz* suggest it has a different function from other *Grl*s.

The lack of detection of transcripts for most *Grl*s in the FlyAtlas 2.0 suggested that these genes might have highly restricted cellular expression patterns. Given the structural similarity of Grls to Grs, we examined their expression in an RNA-seq dataset of the major taste organ (labellum; a tissue not specifically represented in the FlyAtlas 2.0) (*Dweck et al., 2021*). *D. melanogaster Gr* genes display a wide range of expression levels in the labellar transcriptome, in part reflecting the breadth of expression in different classes of taste neurons. For example, *Gr66a* and *Gr64f* – broadly expressed

markers for 'bitter/aversive' and 'sweet/appetitive' neuronal populations, respectively (*Freeman and Dahanukar, 2015*) – are detected at comparatively high levels (>5 FPKM) (*Figure 3F*). By contrast, many receptors expressed in subsets of these major neuron types (e.g., *Gr22e* for bitter and *Gr61a* for sweet; *Freeman and Dahanukar, 2015*) are expressed at much lower levels (~1 FPKM). Similar to this latter type of *Gr*, transcripts for four *Grl*s were detected at >0.5 FPKM: *GrlHz*, *Grl62c*, *Grl62a*, and *Grl36a* (*Figure 3F*). Importantly, within the Fly Cell Atlas dataset of the proboscis and maxillary palp (*Li et al., 2022*), three of these were specifically expressed in the cluster of cells corresponding to *Gr66a*-expressing bitter/aversive neurons (*Figure 3G*). The fourth, *GrlHz*, was very sparsely expressed in non-neuronal cell types in this tissue, including hemocytes (data not shown; *Li et al., 2022*). None of the other six *Grl*s were detectable in this dataset, consistent with their lower expression in the labellar bulk RNA-seq transcriptome (*Figure 3F*). Moreover, no *Grl* (except the broadly expressed *GrlHz*) was detectably expressed in other chemosensory tissue transcriptomes (leg, wing, or antenna) (data not shown; *Li et al., 2022*; *Menuz et al., 2014*). These observations raise the possibility that at least three Grls (Grl36a, Grl62a, and Grl62c) are chemosensory receptors for aversive stimuli.

## A hypothesis for the evolution of the 7TMIC superfamily

Two hypotheses could explain the similarities between well-established 7TMICs and the candidate homologs described in this work: homology (i.e., shared ancestry), and thus the existence of a unified 7TMIC superfamily, or convergent evolution of the 7TMIC structure. We discuss the latter possibility in the following section. Here, we consider a detailed hypothesis of a 7TMIC superfamily of single evolutionary origin. Because confident multiprotein alignment of all members was impossible, we used the same all-to-all graph-based approach as for insect Grls to generate a sequence similarity network, and families were identified as clusters in a 2D projection (*Figure 4A*). We used the gross connectivity of clusters, and the presence or (putative) absence of these proteins across taxa (*Figure 4B*), to make inferences about the ancestry of these proteins.

In the sequence similarity network, clusters of Ors, Grs, and non-insect animal GRLs (excluding GrlHz) were closely located or intermingled, while insect Grl clusters were more distantly located from this grouping (*Figure 4A*). GrlHz formed a distinct cluster, but this connects only with the Or/Gr/Grl clusters (and not plant DUF3537 or PHTF clusters) (*Figure 4A*), suggesting that it descended from a Gr-like ancestor. Given that GrlHz was not detected outside of Holozoa (*Figure 4B*), the simplest hypothesis is that an ancestral holozoan had a 7TMIC gene that duplicated to produce an ancestral GrlHz and an ancestral Gr (*Figure 4C*). The diversity of Ors, Grs, and Grls would then have resulted from taxon-specific diversification of a single, holozoan branch of a hypothetical 7TMIC superfamily (*Figure 4C*).

The plant DUF3537 protein cluster was relatively well connected to the Or/Gr/Grl clusters (*Figure 4A*), consistent with the previously recognized sequence similarity between DUF3537 and Grs, which supported their proposed shared ancestry (*Benton, 2015*; *Benton et al., 2020*). If this is correct, a DUF3537/Or/Gr/Grl ancestor must have been present in a common ancestor of plants (part of Diaphoretickes) and animals (part of Amorphea) (*Figure 4B–C*). Unicellular eukaryotic 7TMICs were dispersed between Or/Gr/Grl and DUF3537 proteins (*Figure 4A*); the simplest hypothesis is that these are related to other 7TMICs in accordance with their species' taxonomy (e.g., SAR [stremenopiles, alveolates, and Rhizaria] 7TMICs are more closely related to plant DUF3537 proteins than to animal Grs). Alternatively, the generally sparse conservation of unicellular eukaryotic 7TMICs might indicate horizontal gene transfer(s).

Finally, PHTF also forms a separate cluster (*Figure 4A*), and its broad taxonomic representation argues that the *PHTF* ancestral gene must also have been present in a common Amorphea-Diaphoretickes ancestor (*Figure 4B–C*). If there was a single ancestral 7TMIC, we hypothesize that this gene must have duplicated in a common eukaryotic ancestor to produce the distinct PHTF and Or/Gr/Grl/DUF3537 lineages (*Figure 4C*).

## Concluding remarks

Exploiting recent advances in protein structure predictions, we have used a tertiary structure-based screening approach to identify new candidate members of the 7TMIC superfamily. While the founder members of this superfamily, insect Ors and Grs, were thought for many years to define an invertebrate-specific protein family (*Benton, 2006*; *Robertson et al., 2003*), there is now substantial

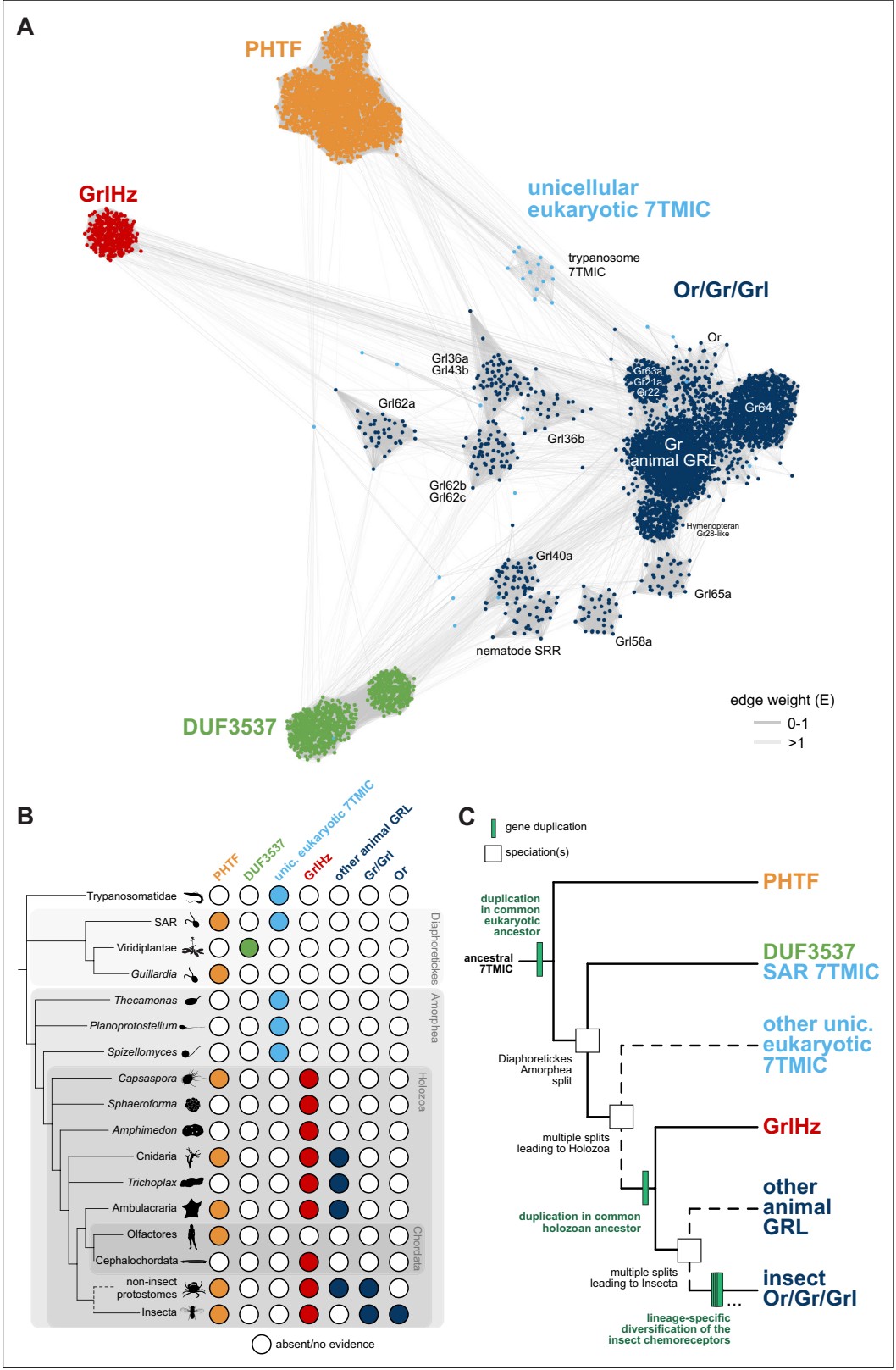

**Figure 4.** A hypothesis for the evolution of the seven transmembrane domain ion channel (7TMIC) superfamily. (**A**) Sequence similarity network of the 7TMIC superfamily, generated using the same odorant receptors (Ors) and gustatory receptors (Grs) from *Figure 3B*, unicellular eukaryotic Grls from *Benton et al., 2020*, and sequence databases assembled using the following query sequences: *N. vectensis* GRL1, *D. melanogaster* Grls and Phtf,

*Figure 4 continued*

*H. sapiens* PHTF1 and PHTF2, *Arabidopsis thaliana* Domain of Unknown Function (DUF) 3537, *C. elegans* SRRs and trypanosome GRLs. The network was generated and visualized as in **Figure 3B**. The graph splitting tree is visualized in **Figure 4—figure supplement 1**. (**B**) Presence and absence of 7TMICs across taxa: 'other animal GRL' refers to GRLs in non-insect animal species previously identified by primary sequence similarity (**Benton, 2015**; **Robertson, 2015**; **Saina et al., 2015**) and nematode SRRs. The dashed branch represents several collapsed paraphyletic clades. (**C**) Model of 7TMIC superfamily evolution. The dashed branches represent several collapsed paraphyletic clades and speciation events. The trypanosome 7TMICs are unplaced due to the currently unresolved taxonomy of trypanosomes (**Burki et al., 2020**).

The online version of this article includes the following source data and figure supplement(s) for figure 4:

**Source data 1.** FASTA file containing the amino acid sequences used in the network and graph splitting analysis of eukaryotic seven transmembrane domain ion channels (7TMICs).

**Source data 2.** Tab delimited text file containing the sequence similarity network of eukaryotic seven transmembrane domain ion channels (7TMICs).

**Source data 3.** Tab delimited text file containing the annotation for the sequence similarity network of eukaryotic seven transmembrane domain ion channels (7TMICs).

**Source data 4.** Newick tree file containing the graph splitting tree of eukaryotic seven transmembrane domain ion channels (7TMICs), derived from the sequence similarity network by gs2.

**Figure supplement 1.** Graph splitting tree for the proposed seven transmembrane domain ion channel (7TMIC) superfamily.

---

evidence that these proteins originated in a eukaryotic common ancestor. We also counter previous assumptions that 7TMICs were completely lost in Chordata, through discovery of two lineages within this superfamily: PHTF and GrlHz. Finally, we have identified many previously overlooked putative chemosensory receptors in *D. melanogaster* (and related flies).

Two important issues remain open. First, are all of the candidate 7TMICs homologous, or does shared tertiary structure reflect convergent evolution in protein folding in at least some cases? Doubts about homology stem, reasonably, from the extreme sequence divergence between 7TMICs to beyond the twilight zone of sequence similarity (**Rost, 1999**). However, sequence divergence over many millions of years of accumulated amino acid substitutions is well appreciated in this superfamily (e.g., pairs of *D. melanogaster* Grs can display as little as 8% amino acid identity; **Robertson et al., 2003**). Thus, sequence dissimilarity alone is not compelling evidence for structural convergence. Examples of convergent evolution of tertiary protein structures have been described (**Alva et al., 2010**; **Alva et al., 2015**; **Tomii et al., 2012**) but the vast majority of these are small protein domains or motifs, some of which might represent relics of the evolution of proteins from short peptide ancestors (**Alva et al., 2015**; **Lupas et al., 2001**). The core of the 7TMIC fold is >300 amino acids, and the question of homology or convergence is most akin to the unresolved, long-standing debate regarding the evolution of the 7TM G protein-coupled receptor fold of type I and type II rhodopsins (**Mackin et al., 2014**; **Rozenberg et al., 2021**). In the case of 7TMICs, if PHTFs and other family members are not homologous, their taxonomic representation indicates that structural convergence must have occurred in a eukaryotic common ancestor. While it might be impossible to definitively distinguish homology from convergence, both hypotheses have interesting implications for this protein fold: convergent evolution of at least some 7TMICs from several distinct origins would argue that the fold is an energetically favorable packing of seven TMs; if the superfamily had a single origin, this would further highlight the remarkable potential for sequence diversification while maintaining a common tertiary structure (**Schaeffer and Daggett, 2011**).

Second, what are the biological roles of different 7TMICs? One aspect of this question pertains to their mechanism of action, that is, whether they assemble in multimeric complexes to form ligand-gated ion channels, similar to insect Ors and Grs. The apparent presence of an anchor domain in all 7TMICs, where most inter-subunit contacts occur in Ors (**Butterwick et al., 2018**; **Del Mármol et al., 2021**), raises the possibility that complex formation is a common biochemical property. Whether they function as ligand-gated ion channels is not necessarily trivial to answer. Even for insect Grs – for which abundant evidence exists for their in vivo requirement in tastant-evoked neuronal activity (**Chen and Dahanukar, 2020**) – definitive demonstration of their chemical ligand-gated ion conduction properties has (with rare exceptions, e.g., **Morinaga et al., 2022**) been elusive. For PHTF or GrlHz proteins,

for example, it is currently difficult to anticipate what might be relevant ligands and we cannot exclude that they have a completely different type of biological activity. Nevertheless, available expression data points to roles of different proteins in specific, but diverse cell types, including chemosensory neurons, (developing) spermatocytes, and muscle. The discoveries in this work should stimulate interest in an even broader community of researchers to understand the evolution and biology of 7TMICs.

## Methods

### 7TMIC candidate homolog identification

Structural screens for candidate 7TMIC homologs were performed with the AF-DB search tool on the Dali server (http://ekhidna2.biocenter.helsinki.fi/dali/; *Holm, 2022*) using as query the *A. bakeri* Orco structure (PDB 6C70-A) (*Butterwick et al., 2018*). As of December 2022, this server permitted screening of the structural proteome of 47 phylogenetically diverse species. Proteins whose structural models had a Z-score >10 were retained for further analysis. Candidate homologs from these screens were assessed first by using these as queries in Dali AF-DB searches of the *D. melanogaster* proteome to ensure Ors and Grs were the best 'reverse' hits, and subsequently for secondary structural features using DeepTMHMM (https://dtu.biolib.com/DeepTMHMM/) (*Hallgren et al., 2022*) and Phobius (https://phobius.sbc.su.se/) (*Käll et al., 2007*). Of the newly identified *D. melanogaster* Grls, we note that three were initially classified as being members of the *Gr* repertoire (Grl36a (Gr36d), Grl43a (Gr43b), and Grl65a (Gr65a), but later excluded (Flybase [flybase.org/] and [*Robertson et al., 2003*])). We also contrast the term 'Grl', referring to the proteins in insects (following nomenclature conventions of *D. melanogaster* [Flybase]) with 'GRL', referring to proteins in other animals and more distant eukaryotes; the same acronym does not reflect a monophyletic origin. To identify sequences of candidate homologs from other species that were not screened with Dali AF-DB, PSI-BLAST searches against the NCBI refseq_protein database were performed, using the query sequences indicated in each figure and dataset. PSI-BLAST was run with an expected threshold of 1E-10 until convergence. BLASTP searches for Gr36/59 homologs were performed more permissively, using an expected threshold of 0.05. All sequences analyzed in this work are provided in *Source data 4*.

### Structure predictions and analysis

AlphaFold2 protein models (*Jumper et al., 2021*; *Varadi et al., 2022*) were downloaded from the AlphaFold Protein Structure Database (alphafold.ebi.ac.uk; release July 2022). For proteins for which structural predictions were not already available, we generated AlphaFold2 models using ColabFold (*Mirdita et al., 2022*). Positive and negative control protein structures were downloaded from the RCSB Protein Data Bank (PDB codes are indicated in *Table 1*). Pairwise structural similarities of protein models were quantitatively assessed with Dali (*Holm, 2022*) and TM-align (https://zhanggroup.org/TM-align/) (*Zhang and Skolnick, 2005*). Proteins were aligned to the same coordinate space with Coot (https://www2.mrc-lmb.cam.ac.uk/personal/pemsley/coot/) (*Emsley et al., 2010*) and visualized in PyMol v2.5.4. All models analyzed in this work are provided in *Source data 1*.

### Phylogenetic and network analyses

Sequence databases assembled using PSI-BLAST (see above) were first curated in a semi-automated pipeline. First, sequences annotated as 'partial' or 'low quality', or that contained ambiguous sequence characters (e.g., X), were removed. CD-HIT (http://cd-hit.org) (*Fu et al., 2012*; *Li and Godzik, 2006*) was used to cluster redundant sequences (100% amino acid identity). Using Phobius TM domain predictions, we removed sequences with fewer than four TMs (this number was chosen to allow for the different sensitivity of Phobius compared to DeepTMHMM). In the final PHTF database, we manually excluded a single sequence as a spurious hit (*B. floridae* XP_035670545.1, zinc transporter ZIP10-like); this sequence sorted independently in first-pass phylogenetic analyses (via FastTree2; *Price et al., 2010*), and a search via InterPro (ebi.ac.uk/interpro/) revealed that it had no obvious similarity to the other proposed homologs. The database of Gr39/Gr59 homologs was manually curated due to its relatively small size and accurate automatic annotation by RefSeq; here, we excluded BLAST hits not annotated as Grs, and visually inspected a sequence alignment for good alignment.

To reduce the large curated sequence databases to a size that could be locally analyzed by both maximum likelihood and Bayesian phylogenetic methods, CD-Hit was used to cluster sequences by

70–90% sequence identity, using the longest sequence as the representative for phylogenetic analyses. The clustering used to generate each phylogeny is indicated in the corresponding figure legend.

We took two separate approaches to infer ancestry. In initial analyses, when comparing insect Grls to Ors/Grs/non-insect GRLs (*Figure 3B*) or for the entire 7TMIC superfamily (*Figure 4A*), we observed that extremely low sequence similarity severely constrained our ability to generate meaningful multiple sequence alignments (data not shown). We therefore generated all-to-all sequence similarity networks using MMSeqs2 and inferred phylogenies from these networks by the graph splitting method (both implemented in gs2) (*Matsui and Iwasaki, 2020*). MMSeqs2, as implemented in gs2, employs high sensitivity to sequence similarity and is thus capable of networking non-homologous sequences via spurious sequence identity, should it be present. This method does not distinguish between homology and convergence. Rather, the purpose of this analysis was to make inferences about relatedness under the assumption that all sequences are homologous. In the networks, edge weights are E-values from MMSeqs2. In the graph splitting trees, branch support values were generated by the edge perturbation method (1000 replicates) with a transfer bootstrap expectation (*Lemoine et al., 2018*). For visualization of sequence similarity networks, recursive, same-to-same sequence comparisons (resulting in an E-value of 0) were removed using an R script.

For all other trees, multiple sequence alignments were generated by MAFFT. We made no a priori assumptions about the alignment, so used default settings. Phylogenetic trees were inferred by maximum likelihood and Bayesian methods. Maximum likelihood trees were generated by IQ-TREE (*Minh et al., 2020*), using the best model selected for each analysis by ModelFinder (*Kalyaana-moorthy et al., 2017*) according to the Bayesian information criterion, and with bootstrapping by UFBoot2 (1000 replicates) (*Hoang et al., 2018*). Bayesian trees were generated by MrBayes (*Ronquist and Huelsenbeck, 2003*) using a mixed amino acid substitution model (Markov chain Monte Carlo analyses run until standard deviation of split frequencies <0.05, with 25% burn in). To generate the most parsimonious hypotheses of protein evolution, we used NOTUNG (*Chen et al., 2000*) to rearrange poorly supported branches and resolve polytomies in a species tree-aware fashion (i.e., favoring speciation to gene duplication/horizontal gene transfer in poorly supported branches and polytomies), using default weights/costs (gene duplication 1.5, transfers 3.0, gene loss 1.0). Branches were eligible for rearrangement at branch support values less than UFboot 95 or posterior probability 0.95. Species trees used for rearrangement were based on the NCBI Taxonomy Common Tree, with polytomies randomly resolved for each analysis using the ape (*Paradis and Schliep, 2019*) and phytools (*Revell, 2012*) packages. Strict consensus trees were generated by comparing the species tree-aware maximum likelihood and Bayesian trees via the consensus function in ape.

The 7TMIC sequence similarity network was visualized and annotated in Cytoscape (*Shannon et al., 2003*). Trees were visualized and annotated in NOTUNG, iTOL (itol.embl.de/) (*Letunic and Bork, 2007*), and Adobe Illustrator. Consensus sequence illustrations were adapted from figures generated by WebLogo (weblogo.berkeley.edu/) (*Crooks et al., 2004*).

## Synteny and intron mapping

The locations of *Grl36a*, *Grl43a*, *Gr36*, and *Gr59c/d* genes in different drosophilids were surveyed using the NCBI Genome Data Viewer (ncbi.nlm.nih.gov/genome/gdv/) (*Rangwala et al., 2021*). Gene intron-exon structures were manually surveyed using publicly available predictions available on RefSeq (via the Genome Data Viewer) and FlyBase, and visualized in SnapGene. The relative positions of introns were assessed via multiple sequence alignment of the protein sequences; for this analysis, we assumed that that entire sequences could be aligned (global alignment), and thus computed the alignment using the G-INS-i (Needleman-Wunsch) option in MAFFT.

## Expression analysis

*H. sapiens PHTF1* and *PHTF2* tissue-specific RNA expression data were obtained from the GTEx Portal (GTEx Analysis Release V8 [dbGaP Accession phs000424.v8.p2; https://gtexportal.org/home/data-sets]). Tissue/life stage-specific RNA expression data of *Phtf* and *Grl* genes in *D. melanogaster* were downloaded from the Fly Atlas 2.0 (https://motif.mvls.gla.ac.uk/FlyAtlas2) (*Krause et al., 2022*) or, for the labellum, from *Dweck et al., 2021*. *D. melanogaster* scRNA-seq data was from the Fly Cell Atlas (*Li et al., 2022*): proboscis/maxillary palp (10× stringent dataset) and testis/seminal vesicle (10×

relaxed dataset), visualized as HVG tSNE or UMAP plots, respectively, in the SCope interface (https://scope.aertslab.org/#/FlyCellAtlas) (*Davie et al., 2018*).

## Acknowledgements

We are very grateful to Julia Santiago for advice on protein structure comparisons and instruction on Coot and Pymol. We acknowledge use of data from the Genotype-Tissue Expression (GTEx) Project, which is supported by the Common Fund of the Office of the Director of the National Institutes of Health, and by NCI, NHGRI, NHLBI, NIDA, NIMH, and NINDS. We thank Roman Arguello, Jamin Letcher, Julia Santiago, and members of the Benton laboratory for comments on the manuscript. Research in RB's laboratory is supported by the University of Lausanne, an ERC Advanced Grant (833548) and the Swiss National Science Foundation. NJH is supported by a Human Frontier Science Program Long-Term Postdoctoral Fellowship (LT-0003/2022L).

## Additional information

### Funding

| Funder | Grant reference number | Author |
|---|---|---|
| H2020 European Research Council | 833548 | Richard Benton |
| Schweizerischer Nationalfonds zur Förderung der Wissenschaftlichen Forschung | 310030B-185377 | Richard Benton |
| Human Frontier Science Program | LT-0003/2022-L | Nathaniel J Himmel |

The funders had no role in study design, data collection and interpretation, or the decision to submit the work for publication.

### Author contributions

Richard Benton, Conceptualization, Data curation, Funding acquisition, Investigation, Methodology, Project administration, Supervision, Validation, Visualization, Writing – original draft, Writing – review and editing, Conceived the project, and performed structural screens/analyses and expression analyses; Nathaniel J Himmel, Conceptualization, Data curation, Funding acquisition, Validation, Investigation, Visualization, Methodology, Writing – original draft, Project administration, Writing – review and editing, Performed sequence-based homolog identification, phylogenetic and network analyses, and gene structure/synteny and protein motif analyses

### Author ORCIDs

Richard Benton (ID) http://orcid.org/0000-0003-4305-8301
Nathaniel J Himmel (ID) http://orcid.org/0000-0001-7876-6960

### Decision letter and Author response

Decision letter https://doi.org/10.7554/eLife.85537.sa1
Author response https://doi.org/10.7554/eLife.85537.sa2

## Additional files

### Supplementary files
• MDAR checklist

• Source data 1. AlphaFold2 models. Models of proteins analyzed in this work, either downloaded from the AlphaFold Protein Structure Database or, where not already available, predicted using the AlphaFold2 algorithm implemented in ColabFold (*Mirdita et al., 2022*). The four-letter code in the filename represents the first letter of the genus and the first three letters of the species (e.g., 'Dmel'

= *D. melanogaster*); species names are given in full in the figures.

• Source data 2. Dali screen search results. Individual text files represent the output of the Dali AF-DB search using *A. bakeri* Or co-receptor (Orco) chain A (PDB 6C70-A) as query and the structural proteome dataset of the indicated species (note the datasets are from version 1 of the AlphaFold Protein Structure Database; subsequent, improved models were used for the pairwise comparisons in *Table 1*). The four-letter codes in the file names and job titles are as described for *Source data 1*.

• Source data 3. Reverse Dali search results. Individual text files represent the output of the Dali AF-DB search using the indicated query candidate seven transmembrane domain ion channels (7TMICs) from *Trypanosoma* (GRL1), *D. melanogaster* (Grls), or *H. sapiens* (PHTF1/2) and the structural proteomic dataset of *D. melanogaster*.

• Source data 4. All uncurated PSI-BLAST sequence databases. Each of the FASTA filenames is formatted as follows (with the exception of the *D. melanogaster* odorant receptor (Or) and gustatory receptor (Gr) sequences, which were collected manually from FlyBase): ProteinFamily-QuerySpecies-QuerySequence.fasta.

### Data availability

All data generated or analysed during this study are included in the manuscript and supporting files.

The following previously published datasets were used:

| Author(s) | Year | Dataset title | Dataset URL | Database and Identifier |
|---|---|---|---|---|
| Krause SA, Overend G, Dow JAT, Leader DP | 2022 | FlyAtlas 2 in 2022: enhancements to the *Drosophila melanogaster* expression atlas | https://motif.mvls.gla.ac.uk/FlyAtlas2 | motif, FlyAtlas2 |
| GTEx Portal | 2021 | The Genotype-Tissue Expression (GTEx) Project | https://www.ncbi.nlm.nih.gov/projects/gap/cgi-bin/study.cgi?study_id=phs000424.v8.p2 | dbGaP, phs000424.v8.p2 |

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
