## [Editor Report]

This article provides fundamental advances to our understanding of the ancestry of insect gustatory and olfactory receptors. It identifies new members of these two related ion channel families in distant species, and the strength of evidence is exceptional. This work will serve as a reference for scientists working on insect olfaction and for those working on molecular evolution

---

## [Decision Letter]

**Decision letter after peer review:**

Thank you for submitting your article "Structural screens identify candidate human homologs of insect chemoreceptors and cryptic *Drosophila* gustatory receptor-like proteins" for consideration by *eLife*. Your article has been reviewed by 3 peer reviewers, and the evaluation has been overseen by Claude Desplan as the Reviewing and Senior Editor. The following individual involved in the review of your submission has agreed to reveal their identity: Hua Yan (Reviewer #1).

The reviewers were very positive about the paper and have discussed their reviews with one another, and the Reviewing Editor has drafted this to help you prepare a revised submission.

Essential revisions:

The three reviewers were extremely positive about the content and presentation of the paper. Please send us as soon as possible an edited version of the paper following the suggestions of the reviewers in order for the paper to be finally accepted.

*Reviewer #1 (Recommendations for the authors):*

I strongly recommend publication, after the authors properly address the following concerns:

Figure 1C: "Note the model contains the extracellular loop 2 (EL2) and intracellular loop 2 (IL2) regions that were not visualized in the cryo-EM structure." Could you explain why? Is this possibly a technical limitation of EM or an inaccurate prediction from AlphaFold2? If the latter, what would be the solution?

In Table 1, what if using AlphaFold2 to analyze negative controls? The same method is necessary for comparison. X-ray and EM data should also be listed to see whether the structures predicted by Alpha2 and X-ray/EM are consistent when analyzing other 7TM proteins.

Better to include one or two vertebrate species beyond mammals/fish in Table 1.

In the Discussion, please give a brief explanation of why AlphaFold2 identified PHTFs and drosophilid GRLs, while trRosetta and RaptorX did not.

The paragraph from lines 268-294 described the expression and potential biological function of PHTF, and finally stated "we suggest they also act as ion channels". However, the evidence (association with E3, cell proliferation, and survival) does not suggest acting as ion channels. If no evidence, can we still say that they belong to 7TMIC? What if later on, we find they are membrane-bound proteins with other functions, but not acting as ion channels?

*Reviewer #3 (Recommendations for the authors):*

1. Figure 1a Caption: Please change to make it clear this is a reprint of data. There is a reference to the original paper, but I would suggest "reprinted from" or "modified from" so it is clear that this is not original data.

2. This is a small point but something worth considering at this stage. "7TMICs" is a tough acronym. I think 7TM GPCRs are usually separated by a space or a hyphen to indicate the separate meaning of the two components. Since work on this channel family is likely to take off after this paper and you will determine the nomenclature now, it is worth considering the name that will be carried with it. Maybe add a space or a hyphen so that the meaning is easy to discern from the abbreviation (ex: 7TM-IC or 7TM IC).

3. Throughout the paper it is helpful to easily be able to find and compare the trypanosome data in various panels. It would be easier to follow if there were a star or something (different color) next to Trypanosoma in 1D so that readers don't need to hunt to relate 1D and 1G, and later to the model in figure 4.

4. Figure 2F: Why are the top two violins and the bottom bar graphs? I'm sure there is a reason, but this seemed like an inconsistency.

5. The two shades of blue used in figure 4 are very difficult to separate when printed. This made it hard to interpret. I would recommend finding higher-contrast colors.

---

## [Author Response]

Reviewer #1 (Recommendations for the authors):I strongly recommend publication, after the authors properly address the following concerns:Figure 1C: "Note the model contains the extracellular loop 2 (EL2) and intracellular loop 2 (IL2) regions that were not visualized in the cryo-EM structure." Could you explain why? Is this possibly a technical limitation of EM or an inaccurate prediction from AlphaFold2? If the latter, what would be the solution?

This is simply a technical limitation of the cryo-EM structure: the Butterwick Nature 2018 paper states that:

“Side-chain density was clearly resolved for most of the Orco channel and 82% of the protein could be accurately modelled, with the exception of the second extracellular loop (Val156–Ile170) and second intracellular loop (Leu244–Asn312)”.

We have expanded the text to clarify this point in the legend.

In Table 1, what if using AlphaFold2 to analyze negative controls? The same method is necessary for comparison. X-ray and EM data should also be listed to see whether the structures predicted by Alpha2 and X-ray/EM are consistent when analyzing other 7TM proteins.

This is a good point; we now present the structural similarity analyses (pairwise Dali and TM-align) for the AlphaFold2 models of the “negative control” proteins in Table 1. The results confirm the analyses with the experimentally-determined structures that these proteins display very low or no similarity to the Orco cryo-EM structure.

Better to include one or two vertebrate species beyond mammals/fish in Table 1.

We assume that the reviewer is referring to the PHTF proteins in Table 1, for which we provide structural similarity scores for the human and *D. melanogaster* homologs. The scores for these different species are very similar, consistent with the high primary sequence similarity of PHTFs across eukaryotes. It is for this reason that we felt that including additional vertebrate PHTFs – which are more closely related to the human proteins than *D. melanogaster* Phtf – in this table would be redundant (and are cognisant that this table is already rather large). We note that the Dali similarity scores of rat, mouse and zebrafish PHTFs to the Orco cryo-EM structure are available in the Supplementary Data, as part of the results of the initial screen.

In the Discussion, please give a brief explanation of why AlphaFold2 identified PHTFs and drosophilid GRLs, while trRosetta and RaptorX did not.

We think that there has been a minor misunderstanding: the identification of PHTFs and drosophilid Grls in this work was made possible because of the availability of AlphaFold2 protein structure database, enabling searching for proteins based upon tertiary structural similarity. To our knowledge, trRosetta and RaptorX have not been used to generate an equivalent database of searchable protein structures.

The paragraph from lines 268-294 described the expression and potential biological function of PHTF, and finally stated "we suggest they also act as ion channels". However, the evidence (association with E3, cell proliferation, and survival) does not suggest acting as ion channels. If no evidence, can we still say that they belong to 7TMIC? What if later on, we find they are membrane-bound proteins with other functions, but not acting as ion channels?

We fully agree that some (or even many) of the 7TMICs we describe might not function as ion channels, and stress that the “7TMIC” name was principally to avoid the cumbersome “Or/Gr/Grl/DUF3537/PHTF” terminology when collectively referring to these proteins. Having said this, the properties of mammalians PHTFs mentioned by the reviewer are not incompatible with an ionotropic function: the association with the E3 ubiquitin ligase is via the cytoplasmic N-terminus that is distinct from the presumed ion channel domain (and several other known ion channels associate with E3 proteins). Moreover, it remains to be determined how directly PHTFs regulate cell proliferation/survival.

Reviewer #3 (Recommendations for the authors):1. Figure 1a Caption: Please change to make it clear this is a reprint of data. There is a reference to the original paper, but I would suggest "reprinted from" or "modified from" so it is clear that this is not original data.

We have indicated that the Orco structure we present is “derived from” PDB 6C70 (Buttewick et al., 2018), which we feel is the most appropriate phrasing, in that we used the structural coordinates of the PDB but visually present it in a new way.

2. This is a small point but something worth considering at this stage. "7TMICs" is a tough acronym. I think 7TM GPCRs are usually separated by a space or a hyphen to indicate the separate meaning of the two components. Since work on this channel family is likely to take off after this paper and you will determine the nomenclature now, it is worth considering the name that will be carried with it. Maybe add a space or a hyphen so that the meaning is easy to discern from the abbreviation (ex: 7TM-IC or 7TM IC).

We have thought quite a lot about the nomenclature in preparing this manuscript, and stress that our proposal of “7TMIC” (which we find can be pronounced “seven-tea-mick” fairly fluently!) should be considered a placeholder name that we use here to avoid the cumbersome “Or/Gr/Grl/DUF3537/PHTF” collective terminology in the text. We would welcome any future revision of this family name, particularly when functional data is obtained. Should the proposed name become the standard in the field, we felt that the reviewer’s alternative proposals (with a hyphen or space) would be liable to inconsistent application of the punctuation, compared to the compact “7TMIC”. Note that we avoided using “7TMC” to prevent confusion with various other TMC protein families, which are unrelated to the proteins studied in this work.

3. Throughout the paper it is helpful to easily be able to find and compare the trypanosome data in various panels. It would be easier to follow if there were a star or something (different color) next to Trypanosoma in 1D so that readers don't need to hunt to relate 1D and 1G, and later to the model in figure 4.

We were not sure why the reviewer would like to highlight the Trypanosoma proteins in particular in Figure 1D (the species names are already indicated, and the corresponding proteins are presented in bold in Figure 1G). This plot (reporting the results of the screen) is the basis for the further analysis of Trypanosome GRLs, PHTFs and insect Grls in the subsequent figures, so it seemed to us to be unnecessarily explicit to specifically point to the Trypanosoma hits. In the model in Figure 4C, we state in the figure legend “The trypanosome 7TMICs are unplaced, due to the currently unresolved taxonomy of trypanosomes (Burki et al., 2020).”; nevertheless, these proteins are labelled in Figure 4A-B to permit their easy location.

4. Figure 2F: Why are the top two violins and the bottom bar graphs? I'm sure there is a reason, but this seemed like an inconsistency.

The different plotting simply reflects the nature of the data available: the human *PHTF1/2* expression data is a compilation of many independent RNA-seq studies (available from the GTEx Portal), while the *D. melanogaster Phtf* expression data is the mean FPKM (± SD) that is available from the Fly Atlas 2.0; the individual values of the three biological replicates of this atlas are not available. (There are of course many other differences between these datasets, not least the range of tissue analyzed).

5. The two shades of blue used in figure 4 are very difficult to separate when printed. This made it hard to interpret. I would recommend finding higher-contrast colors.

We have increased the contrast between these two blues and generated a higher quality image for Figure 4A, so that the contrast is more evident.